# Polymorphisms in Glyoxalase I Gene Are Not Associated with Glyoxalase I Expression in Whole Blood or Markers of Methylglyoxal Stress: The CODAM Study

**DOI:** 10.3390/antiox10020219

**Published:** 2021-02-02

**Authors:** Kim Maasen, Nordin M. J. Hanssen, Carla J. H. van der Kallen, Coen D. A. Stehouwer, Marleen M. J. van Greevenbroek, Casper G. Schalkwijk

**Affiliations:** 1Department of Internal Medicine, CARIM School for Cardiovascular Diseases, Maastricht University Medical Centre, Universiteitssingel 50, 6200 MD Maastricht, The Netherlands; kim.maasen@maastrichtuniversity.nl (K.M.); c.vanderkallen@maastrichtuniversity.nl (C.J.H.v.d.K.); cda.stehouwer@mumc.nl (C.D.A.S.); m.vangreevenbroek@maastrichtuniversity.nl (M.M.J.v.G.); 2Department of Internal and Vascular Medicine, Amsterdam University Medical Centres, Meibergdreef 9, 1105 AZ Amsterdam, The Netherlands; nmj.hanssen@maastrichtuniversity.nl

**Keywords:** glyoxalase 1, methylglyoxal, D-lactate, advanced glycation endproducts, gene expression, mRNA, single nucleotide polymorphism, CODAM, glyoxalase pathway

## Abstract

Glyoxalase 1 (Glo1) is the rate-limiting enzyme in the detoxification of methylglyoxal (MGO) into D-lactate. MGO is a major precursor of advanced glycation endproducts (AGEs), and both are associated with development of age-related diseases. Since genetic variation in *GLO1* may alter the expression and/or the activity of Glo1, we examined the association of nine SNPs in *GLO1* with Glo1 expression and markers of MGO stress (MGO in fasting plasma and after an oral glucose tolerance test, D-lactate in fasting plasma and urine, and MGO-derived AGEs CEL and MG-H1 in fasting plasma and urine). We used data of the Cohort on Diabetes and Atherosclerosis Maastricht (CODAM, *n* = 546, 60 ± 7 y, 25% type 2 diabetes). Outcomes were compared across genotypes using linear regression, adjusted for age, sex, and glucose metabolism status. We found that SNP4 (rs13199033) was associated with Glo1 expression (AA as reference, standardized beta AT = −0.29, *p* = 0.02 and TT = −0.39, *p* = 0.3). Similarly, SNP13 (rs3799703) was associated with Glo1 expression (GG as reference, standardized beta AG = 0.17, *p* = 0.14 and AA = 0.36, *p* = 0.005). After correction for multiple testing these associations were not significant. For the other SNPs, we observed no consistent associations over the different genotypes. Thus, polymorphisms of *GLO1* were not associated with Glo1 expression or markers of MGO stress, suggesting that these SNPs are not functional, although activity/expression might be altered in other tissues.

## 1. Introduction

Glyoxalase 1 (Glo1) is the rate-limiting enzyme in the detoxification of methylglyoxal (MGO) into D-lactate [1]. MGO is a highly reactive compound that is mainly formed during glycolysis and lipid peroxidation. It is a major precursor in the formation of advanced glycation endproducts (AGEs), and both MGO and MGO-derived AGEs are associated with the development of age-related diseases, such as diabetes and its associated complications, cardiovascular disease in particular [2,3,4,5].

Detoxification of MGO via the glyoxalase system limits MGO stress; i.e., results in lower MGO concentrations and less formation of MGO-derived AGEs [6,7]. Inhibition of Glo1 increases MGO accumulation, expression of inflammation and endothelium dysfunction markers, and decreases cellular viability [8,9,10,11]. The glyoxalase system, and in particular its rate-limiting component Glo1, may be a key determinant of interindividual susceptibility to elevated MGO levels in the setting of hyperglycemia and subsequent development of diabetic complications, as several experimental studies have linked dysfunction of Glo1 to a higher prevalence of diabetes [12].

In rats, overexpression of Glo1 decreased diabetes-induced accumulation of MGO and MG-H1, oxidative stress and endothelial dysfunction, and attenuated early renal impairment [10,13]. In humans, low Glo 1 activity was associated with painful diabetic neuropathy [14], plaque rupture (as reviewed in [15]), and coronary artery disease [11]. Interestingly, a recent integrative genomics study revealed *GLO1* as a key regulatory gene in coronary artery disease-related processes [16].

Genetic variation in *GLO1* may alter the expression and/or the activity of Glo1, and may thus represent life-long exposure to a higher or lower detoxification potency and MGO stress [3]. Nine single nucleotide polymorphisms (SNPs) have been identified, that cover the total common variability in *GLO1* [17]. In this explorative study, we examined the association of these nine SNPs with gene expression of *GLO1* in white blood cells. In addition, we studied the associations of the SNPs with severity of MGO-stress as estimated by concentrations of MGO, D-lactate, and MGO-derived AGEs in fasting plasma, with the formation of MGO after an oral glucose tolerance test (OGTT), and with concentrations of MGO-derived AGEs and D-lactate in urine. These analyses were done using data from individuals of the Cohort on Diabetes and Atherosclerosis Maastricht (CODAM).

## 2. Materials and Methods

### 2.1. Study Population

Analyses were performed in the CODAM study, which includes 574 individuals with a moderately increased risk for type 2 diabetes and cardiovascular disease as described in detail elsewhere [18]. In short, participants are of Caucasian descent and >40 years of age with one or more of the following characteristics: BMI > 25 kg/m^2^; use of antihypertensive medication; positive family history of type 2 diabetes; postprandial blood glucose level > 6.0 mmol/L; and history of gestational diabetes and/or glycosuria. All 574 participants were extensively characterized at baseline between 1999 and 2002. At the follow-up examination between 2006 and 2009, with a median follow-up period of 7.0 years (IQR 6.9–7.1), the measurements were repeated in 491 individuals.

For the present evaluation, we used data at baseline, except for the *GLO1* mRNA data, which was only available at follow-up. Individuals who had missing DNA samples and therefore missing data for all genotyped SNPs (*n* = 6) were excluded from the current analyses. Additionally, individuals with missing data on fasting plasma concentrations of MGO, D-lactate, or MGO-derived AGEs were excluded (*n* = 22). Participants with missing data on any of the other outcomes (*GLO1* mRNA, iAUC of MGO, urinary AGEs, and urinary D-lactate) and main independent variables (the nine SNPs) were not excluded to maximize the sample size for each analysis. Hence, the maximal study population consisted of 546 participants (see flowchart Appendix A). The study was approved by the Medical Ethics Committee of the Maastricht University Medical Centre. All participants gave written informed consent.

### 2.2. Single Nucleotide Polymorphism Selection and Genotyping

SNP selection was based on *GLO1* (Gene accession number NC_000006.11 and NC_000006.12), including 3000 base pairs downstream and upstream as previously described [17]. Common SNPs (minor allele frequency (MAF) > 5%, *n* = 28) were selected using HapMap and Haploview. By genotyping nine tag SNPs, these 28 common SNPs were captured at *r*^2^ more than 0.8 in linkage disequilibrium, and therefore, total common genetic variability of *GLO1* was covered (Figure 1) [17]. For genotyping, DNA was extracted from peripheral blood samples according to standard procedures. *GLO1* SNPs were genotyped using the ABI PRISM 7900HT sequence detection system (Applied Biosystems, Foster City, CA, USA). Hardy–Weinberg equilibrium was previously assessed using a χ^2^ test [17]. 

### 2.3. GLO1 mRNA Expression in Whole Blood

To ensure adequate stabilization of mRNA, whole blood was collected in PAX gene tubes (Qiagen) and stored at −80 °C until further use. RNA was isolated in an automated fashion with the QIAcube, according to the manufacturer’s instructions. RNA quantity and quality was assessed in a subset of samples (*n* = 216) to ensure the quality of RNA was adequate for qPCR (RIN was >5 in 99.5%). Next, 500 ng RNA per sample was used to synthesize cDNA using miScript, according to the manufacturer’s instructions. Next, qPCR of GLO1 was performed. HRPT1 and WHKY1 were used as reference genes. Primer sequences were for GLO1-forward, 5′-GGTTTGAAGAACTGGGAGTCAAA-3′ and for GLO1-reverse, 5′-ATCCAGTAGCCATCAGGATCTTG-3′; for YWHA-forward, 5′-CGTTACTTGGCTGAGGTTGC-3′ and for YWHA-reverse, 5′-TGCTTGTTGTGACTGATCGAC-3′; and for HRPT1-forward, 5’-AAG-AAT-GTC-TTG-ATT-GTG-GAA-GA-3′ and HRPT1-reverse, 5’-ACC-TTG-ACC-ATC-TTT-GGA-TTA-3’.

### 2.4. Markers of MGO Stress 

Concentrations of MGO, D-lactate, and free N_ε_-(1-carboxyethyl)lysine (CEL), free N_δ_-(5-hydro-5-methyl-4-imidazolon-2-yl)-ornithine (MG-H1), and protein-bound CEL, were measured in fasting plasma samples, using ultra-performance liquid chromatography-tandem mass spectrometry (UPLC–MS/MS) as previously described [19,20,21]. Concentrations of free CEL, free MG-H1, and D-lactate were measured in fasting urine samples, as previously described [21,22]. To correct for urine volume, creatinine concentration in urine was analyzed based on the Jaffé reaction method [23], and CEL, MG-H1, and D-lactate concentrations were expressed as nmol/mmol creatinine. The intra- and interassay variation was between 3.6 and 4.3% for MGO, between 2.9 and 5.2% for D-lactate, between 2.8 and 7.1% for free CEL and MG-H1, and between 4.8 and 9.7% for protein-bound CEL in plasma. The intra- and interassay variations of D-lactate, free CEL, and free MG-H1 in urine were between 3.7 and 7.0%. 

### 2.5. iAUC of MGO after an OGTT 

Individuals underwent a standard 75-g OGTT and venous blood samples were obtained prior to and at 30, 60, and 120 min after the glucose load. Individuals with known type 2 diabetes or with fasting glucose levels >8.5 mmol/L were excluded from undergoing an OGTT. Concentrations of MGO were quantified in plasma samples collected during the OGTT using UPLC–MS/MS. The area under the curve for the OGTT levels of MGO was calculated according to the trapezoidal method [24], where baseline (fasting) levels were subtracted from each individual data point to specify the post-glucose load increases. These data are referred to as iAUC.

### 2.6. Definition of Glucose Metabolism Status 

Individuals’ glucose metabolism status was ascertained using fasting and 2-h postload glucose concentrations during the OGTT, according to the World Health Organization criteria, as described in detail elsewhere [18]. Briefly, individuals were classified as having normal glucose metabolism when they had normal fasting (<6.1 mmol/L) and 2-h postload (<7.8 mmol/L) glucose concentrations. Individuals with impaired fasting glucose (6.1–7.0 mmol/L), impaired 2-h postload glucose levels (7.8–11.1 mmol/L), or both were classified as having impaired glucose metabolism. When individuals had high fasting plasma glucose levels (≥7.0 mmol/L) and/or high 2-h postload glucose levels (≥11.1 mmol/L) or when they used glucose-lowering medication or insulin, they were classified as having type 2 diabetes.

### 2.7. White Blood Cell Composition

Proportions of white blood cells were estimated based on DNA methylation data, using the EpiDISH package through the algorithm CIBESORT (Teschendorff, Breeze, Zheng, and Beck, 2017).

### 2.8. Statistical Analysis

First, reciprocal relationships among all outcome variables (*GLO1* mRNA, plasma fasting concentrations of MGO, D-lactate, free CEL, free MG-H1, protein-bound CEL, iAUC MGO, and urinary concentrations of free CEL, free MG-H1, and D-lactate) were examined using Spearman’s correlation coefficient. Skewed outcome variables (i.e., all except fasting plasma MGO) were ln-transformed prior to further analyses.

In the main analyses, outcome variables were compared across the genotypes of the nine *GLO1* SNPs using multiple linear regression analyses. This was done using additive models with the major homozygous genotype as a reference and the other two genotypes as dummy variables. Outcome variables were standardized ((participants’ value—population mean)/SD) to allow direct comparison of the strength of associations. The results of the linear models are displayed after adjustment for age (years), sex (men/women), and glucose metabolism status (impaired glucose metabolism and type 2 diabetes as dummy variables with normal glucose metabolism as the reference category).

As a sensitivity analysis, the associations with *GLO1* mRNA expression as an outcome were additionally adjusted for white blood cell type composition, because Glo1 expression can vary between cell types. For this, proportions were added as covariates to the model for all but one of the cell types, to improve model stability because cell proportions sum to 1 (B-cells, NK-cells, CD4T-cells CD8T-cells, monocytes, neutrophils and eosinophils, and dropping granulocytes). Cell composition data was available for 162 individuals. 

To correct for multiple testing, false discovery rate- (FDR-) based values were calculated for each outcome, using the *p*-values of the fully adjusted model (18 *p*-values: 9 SNPs × 2 tests per SNP) [25]. The FDR analysis was performed separately for each outcome, because FDR assumes that the *p*-values corresponding to the null hypothesis tests are independent, and in this study the outcome variables are markers of the same pathway and thus not independent. Therefore, calculation of FDR based values for all outcomes simultaneously would likely lead to overcorrection. Significance of the associations was assessed by a 0.05 threshold of value (q < 0.05). 

All models were checked for the assumptions of linearity, normal distribution of residuals, homoscedasticity, and multicollinearity. In the model with iAUC MGO as the outcome variable, the assumption of normality was violated, and therefore as a sensitivity analysis the association between low/high iAUC MGO concentrations (dichotomized by performing a median split) and the nine *GLO1* SNPs was examined using logistic regression. In the analysis between SNP16 and *GLO1* mRNA, one influential outlier was excluded because Cook’s distance was 1.1. Reported betas are after exclusion of the influential outlier. All analyses were performed using the Statistical Package for Social Sciences (SPSS, version 24.0) and statistical significance was set at *p* < 0.05.

## 3. Results

General characteristics for the study population are shown in Appendix A. Excluded individuals were slightly older and more often males. They more often had type 2 diabetes, were less physically active and smoked less (data not shown). All genotyped SNPs were in the Hardy–Weinberg equilibrium. 

### 3.1. Correlations between GLO1 mRNA Expression and Markers of MGO Stress

Reciprocal correlations among all outcomes are shown in Table 1. Glo1 mRNA expression was positively correlated with plasma and urinary concentrations of D-lactate, the product of MGO detoxification (rho = 0.2). *GLO1* mRNA expression was also positively correlated with concentrations of its substrate, MGO, in fasting plasma (rho = 0.2), and iAUC MGO and MGO-derived AGEs in urine (rho = 0.1). Notably, *GLO1* mRNA was measured in samples collected at follow-up, seven years after measurements of the other outcomes. Most other associations were relatively weak (rho ranging from −0.2 to 0.3). We observed a strong reciprocal correlation between the free forms of the MGO-derived AGEs, CEL and MG-H1, measured in plasma (rho = 0.7), as well as between CEL and MG-H1 measured in urine (rho = 0.6). Additionally, we observed a strong correlation between free CEL measured in plasma and in urine, and the same holds true for free MG-H1 in plasma and urine (both rho = 0.7). 

### 3.2. Association between GLO1 Polymorphisms and GLO1 mRNA Expression in Whole Blood

The mean *GLO1* mRNA expression for each genotype of the nine SNPs is shown in Appendix A. After adjustment for age, sex, and glucose metabolism status, carriers of the AT or the TT genotype of SNP4 (rs13199033) had lower *GLO1* mRNA expression than those with the AA genotype (AA as reference, beta AT = −0.29, *p* = 0.02 and beta TT = −0.39, *p* = 0.30; Table 2). The effect was additive, but not statistically significant for the TT genotype, likely due to the small sample size (*n* = 7).

Carriers of the AG or the AA genotype of SNP13 (rs3799703) had higher *GLO1* mRNA expression than those with the GG genotype, in an additive manner (beta AG = 0.17, *p* = 0.14 and beta AA = 0.36, *p* = 0.005; Table 2).

After additional adjustment for white blood cell composition in a subset of the population with cell composition data available (*n* = 162), the strength of the associations remained similar, although none of the associations was statistically significant, likely due to low power.

When we corrected these analyses for multiple testing using FDR, none of these associations remained statistically significant.

### 3.3. Association between GLO1 Polymorphisms and Markers of MGO Stress

The mean of each marker of MGO stress for each genotype of the nine SNPs is shown in Appendix A. In line with their higher *GLO1* mRNA expression, carriers of the AG or the AA genotype of SNP13 (rs3799703) had higher concentrations of plasma fasting D-lactate than those with the GG genotype, albeit not additive and only statistically significant for the AG genotype (beta AG = 0.29, *p* = 0.004 and beta AA = 0.08, *p* = 0.50; Table 3). Carriers of the GT or the GG genotype of SNP18 (rs2736654) had lower concentrations of fasting plasma MGO than those of the TT genotype, but this was only statistically significant for the GT genotype (beta GT = −0.20, *p* = 0.04 and beta GG beta = −0.16, *p* = 0.20; Table 3). Additionally, carriers of the AG or the GG genotype of SNP49 (rs1049346) had lower free CEL concentrations in plasma than those of the AA genotype, but this was only statistically significant for the AG genotype (beta AG = −0.24, *p* = 0.02 and beta GG = −0.15, *p* = 0.23; Table 3). In the sensitivity analyses where the dichotomized variable iAUC MGO was used as the outcome variable, results were similar to the linear regression, with no statistically significant associations (Appendix A). 

After correction for multiple testing using FDR none of these associations remained statistically significant.

## 4. Discussion

In the present study we examined if polymorphisms in *GLO1* were associated with *GLO1* mRNA expression in whole blood and/or with MGO and MGO-derived AGEs. We hypothesized that if these polymorphisms are functional, they can serve as a proxy of a life-long exposure to enhanced or decreased MGO stress, due to altered detoxification potency for MGO.

We observed associations of SNP4, SNP13, SNP18, and SNP49 with various outcomes, but after correction for multiple testing with FDR none of these associations remained statistically significant. Thus, these findings are possibly chance findings, but notwithstanding the observed associations could be of interest. 

There are several possible explanations for the apparent null findings after correction for multiple testing. First, the SNPs of *GLO1* could be non-functional, and have no effect on the activity of the Glo1 enzyme. Second, the effects of these SNPs could be compensated by other mechanisms, for example upregulation of enzyme transcription/activity of Glo1 or other relevant enzymes [26,27,28,29]. Third, the -possibly mild- effects of these SNPs could be overshadowed by other conditions that affect Glo1 transcription/activity, such as hyperglycemia, inflammation, hypoxia, or oxidative stress [3]. Fourth, it could be that markers of MGO stress and Glo1 expression in white blood cells do not reflect altered enzyme activity/function in other cells and/or tissues, and results might be different if cellular Glo1 activity and/or AGE accumulation in different tissues were evaluated as outcomes. Last, correction for multiple testing by FDR could be too strict for these analyses. FDR assumes that *p*-values are independent. Since our outcomes are all markers of the glyoxalase pathway, we calculated the FDR-based *p*-values separately for each outcome. However, due to the dependent nature of genetic data, SNPs in linkage disequilibrium are also correlated to some degree, so the FDR analyses, using *p*-values of all nine SNPs with two genotypes each, might have been too stringent. Thus, the current, explorative evaluation might be underpowered due to the relative small sample size, and replication in a larger, independent cohort is warranted.

In our current analyses, carriers of the GT or the GG genotype of SNP18 (rs2736654) had lower concentrations of plasma MGO than those of the TT genotype, although only significant (nominal *p*-value) for the GT genotype. Additionally, carriers of the AG or the GG genotype of SNP49 (rs1049346) had lower plasma CEL concentrations than those of the AA genotype, which was significant for the AG genotype (nominal *p* value). In line, we observed a trend of higher *GLO1* mRNA expression and urinary D-lactate for carriers of these genotypes. SNP18 (rs2736654) is a nonsynonymous SNP located in exon 4. SNP49 (rs1049346) is located in the 5′ untranslated region (UTR) and may be involved in disrupting the regulation of expression, resulting in a lower enzyme concentration [30]. Studies on associations of these SNPs with markers of the glyoxalase pathway are inconsistent. Some studies in human on SNP18 reported that carriers of the AA (i.e., TT) genotype had the highest Glo1 activity in blood [31,32], whereas others reported only a trend towards higher Glo1 activity [30], or no association [33]. One of these studies additionally reported higher Glo1 activity and lower AGE concentrations in post-mortem brains of carriers of the AA genotype, in healthy individuals but not in autistic individuals [31]. The same study reported no differences in *GLO1* mRNA expression or protein levels, whereas others reported higher *GLO1* mRNA levels for carriers of the AA genotype [32], which is in line with the trend that we observed. This would suggest that carriers of the AA (i.e., TT) genotype might have been exposed to a smaller lifelong MGO stress. However, contrary to these data in human, in immortalized lymphoblastoid cells the AA genotype showed lower Glo1 activity and higher MGO concentrations [34]. Interestingly, in the present evaluation we also observed higher MGO concentrations for carriers of the AA genotype, despite the higher *GLO1* mRNA for this genotype. Considering that SNP18 is located on an exon, it is plausible that SNP18 alters the functionality of Glo1, resulting in altered MGO concentrations.

A previous human study on SNP49 reported that Caucasian carriers of CT (i.e., AG) and TT (i.e., AA) genotypes had a lower Glo1 enzyme activity compared to those of the CC genotype (i.e., GG), measured in whole blood samples [30]. This is in line with the trend of lower *GLO1* mRNA expression for the AG and the AA genotype in the current evaluation. Considering that SNP49 is located in the 5′UTR, it is conceivable that this *GLO1* promoter polymorphism has a functional influence on transcriptional regulation. However, in contrast, in a human cell line the -7T promoter of SNP49 was found to have a higher activity than the -7C promoter [35].

We observed an additive effect on *GLO1* mRNA expression in carriers of the T allele at SNP4 (rs13199033) (AA > AT > TT). In line, there was a trend of higher fasting plasma MGO concentrations in these individuals (AA < AT < TT). SNP4 (rs13199033) is located in the 3′UTR, whereas the other SNPs are located in introns [17]. Although SNP4 has not been reported as functional SNP, our association with *GLO1* mRNA is in line with its location in the 3′UTR.

To date, little is known about the association between these nine SNPs and diabetic complications. One study reported an increased prevalence of peripheral neuropathy, but not nephropathy or retinopathy, for carriers of the CC genotype of SNP18 in type 2 diabetic patients [36]. A study from our group reported no associations of the nine investigated SNPs with vascular complications in this population [17]. Moreover, in this previous study we did not observe an association between these nine SNPs and concentrations of the plasma AGEs, CEL and N_ε_-(carboxymethyl)lysine (CML) [17]. Similarly, we reported only minor associations of the two reported functional SNPs (i.e., SNP18 and SNP49) with plasma concentrations of MGO, free CEL, CML, and MG-H1 and protein-bound CML, CEL, and pentosidine in the same population [37]. Other authors also reported no association between six *GLO1* SNPs and serum CML concentrations [38] and SNP18 and serum MG-H1 [33]. We herein extend these findings to a full panel of markers of MGO stress, including MGO, D-lactate, and MGO-derived AGEs in plasma and urine. This comprehensive analysis shows that indeed these two functional SNPs, and the other studied SNPs, after correction for multiple testing, are not associated with lower or higher concentrations of markers of MGO stress, and thus likely do not play a large role in the detoxification capacity of MGO.

When we explored the reciprocal correlations of all outcome variables, we observed that although Glo1 catalyzes the detoxification of MGO into D-lactate, *GLO1* mRNA expression was not only positively correlated with D-lactate, but also with MGO concentrations. Given the seven year difference between sample collection for measurement of Glo1 expression and measurement of the other outcomes, these correlations have to be interpreted with care, but a possible underlying mechanism is the upregulation of Glo1 as a response to elevated MGO concentrations, via KEAP1 modification and subsequent Nrf1 activation, which regulates Glo1 transcription [39].

We previously showed formation of MGO in plasma after an OGTT, with higher peaks for individuals with prediabetes and type 2 diabetes. These higher peaks of MGO were attributed to an increased formation of MGO from higher blood glucose peaks, but decreased detoxification of MGO by Glo1 in these individuals could also play a role. Our observation of equal genotype frequencies between individuals with normal glucose metabolism, impaired glucose metabolism and type 2 diabetes [17], and the lack of association between *GLO1* SNPs and iAUC of MGO after an OGTT do not support the possibility that impaired detoxification capacity of MGO caused by *GLO1* SNPs are an important contributor to the higher MGO peaks observed after an OGTT.

This study had several strengths. First, we quantified an extensive panel of markers of the glyoxalase pathway and MGO stress, using state-of-the-art techniques. Although these are all markers of the same pathway, they might reflect different (metabolic) entities, as shown by the low reciprocal correlations between the markers. Second, using polymorphisms of *GLO1* as a predictor of the Glo1 enzyme mitigates the limitation of reversed-causality often encountered in cross-sectional studies. The main limitation of this study is its limited power due to the small sample size. Hence we may have missed true positive associations that would be significant in larger study populations (Type II error). Additionally, we measured *GLO1* mRNA expression in white blood cells, but we do not know to what extent this reflects *GLO1* mRNA expression and/or activity in other cells and tissues [28]. In addition, our study only included Caucasian individuals and cannot be extrapolated to different ethnicities. 

## 5. Conclusions

In conclusion, after correction for multiple testing, polymorphisms in *GLO1* are not associated with *GLO1* mRNA expression and markers of MGO stress, in a Dutch cohort of middle-aged to elderly Caucasian individuals with a moderate risk of cardiometabolic diseases. These null findings have to be interpreted with caution because of the chance of Type II error due to the small sample size, and replication in an independent cohort is warranted. 

## Figures and Tables

**Figure 1 antioxidants-10-00219-f001:**
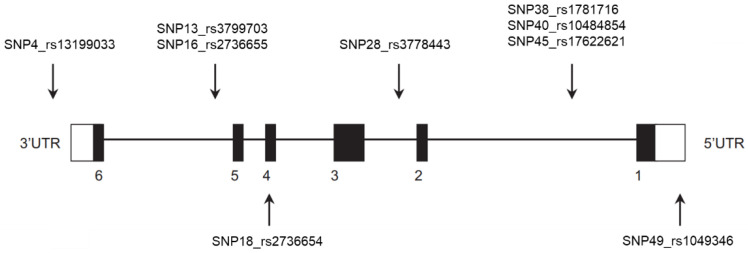
Representation of the human glyoxalase 1 gene. Arrows indicate the approximate locations of the nine single nucleotide polymorphisms (SNPs) genotyped in this study. Black boxes represent exons and untranslated regions (UTRs) of the first and last exon are indicated as white boxes. SNP4 is located on the 3′ untranslated region; SNP18 in exon 4 (nonsynonymous); SNP49 in exon 1 (untranslated region); and all other SNPs are located in introns.

**Table 1 antioxidants-10-00219-t001:**
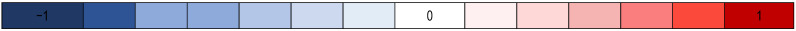
Correlations between *GLO1* mRNA, MGO, D-lactate and AGEs in plasma, and AGEs and D-lactate in urine ^1^.

	Glo1 Expr.	PlasmaMGO and D-Lactate	PlasmaAGEs	UrinaryAGEs and D-Lactate
mRNA	MGO	D-lactate	iAUC MGO	Free CEL	Free MG-H1	PB CEL	Free CEL	Free MG-H1	D-lactate
Glo1 expr.	mRNA	mRNA	0.15 **	0.17 **	0.10 *	0.08	0.08	−0.003	0.12 *	0.11 *	0.21 **
PlasmaMGO andD-lactate	MGO		MGO	0.07	−0.23 **	0.16 **	0.10 *	0.06	0.09 *	−0.01	0.08
D-lactate			D-lactate	0.15 **	0.17 **	0.09 *	0.33 **	0.21 **	0.12 **	0.33 **
iAUC MGO				iAUC MGO	0.05	−0.02	−0.04	0.12*	0.02	0.10 *
PlasmaAGEs	Free CEL					Free CEL	0.69 **	0.10 *	0.67 **	0.45 **	0.05
Free MG-H1						Free MG-H1	0.07	0.45**	0.75 **	0.01
PB CEL							PB CEL	0.04	0.02	0.05
UrinaryAGEs and D-lactate	Free CEL								Free CEL	0.58 **	0.32 **
Free MG-H1									Free MG-H1	0.20 **
D-lactate										D-lactate

^1^ rho = Spearman’s correlation coefficient. * *p* < 0.05, ** *p* < 0.01. Note: all outcomes were measured at baseline (1999–2002), except *GLO1* mRNA expression, which was measured at follow-up (2006–2009). AGE, advanced glycation endproduct; CEL, Nε-(1-carboxyethyl)lysine; Glo1, glyoxalase-1; iAUC MGO, incremental area under the curve of MGO after an OGTT; MGO, methylglyoxal; MG-H1, Nδ-(5-hydro-5-methyl-4-imidazolon-2-yl)-ornithine; OGTT, oral glucose tolerance test; PB, protein-bound.

**Table 2 antioxidants-10-00219-t002:** Association of *GLO1* SNPs with *GLO1* mRNA, plasma MGO and D-lactate, and urinary D-lactate ^1^.

		Glo1 Expression		Plasma						Urine	
		*GLO1* mRNA ^2^		MGO		D-Lactate		MGO iAUC		D-Lactate	
	N	β	*p*	β	*p*	β	*p*	β	*p*	β	*p*
**SNP4**											
AA	423	-	*-*	-	*-*	-	*-*	-	*-*	-	-
AT	102	−0.29	*0.02*	0.07	*0.54*	0.05	*0.62*	0.08	*0.52*	0.11	*0.27*
TT	7	−0.39	*0.30*	0.64	*0.08*	−0.08	*0.83*	−0.37	*0.38*	−0.04	*0.91*
**SNP13**											
GG	148	-	*-*	-	*-*	-	*-*	-	*-*	-	*-*
AG	249	0.17	*0.14*	0.02	*0.85*	0.29	*0.004*	−0.01	*0.91*	0.02	*0.85*
AA	126	0.36	*0.005*	0.05	*0.65*	0.08	*0.50*	0.07	*0.59*	−0.03	*0.76*
**SNP16**											
GG	397	-	*-*	-	*-*	-	*-*	-	*-*	-	*-*
AG	127	0.15	*0.17*	−0.01	*0.89*	0.12	*0.23*	0.09	*0.41*	0.04	*0.64*
AA	11	0.21	*0.46*	−0.11	*0.71*	−0.17	*0.57*	0.04	*0.91*	0.06	*0.85*
**SNP18**											
TT	161	-	*-*	-	*-*	-	*-*	-	*-*	-	*-*
GT	277	−0.07	*0.49*	−0.20	*0.04*	−0.03	*0.78*	−0.03	*0.77*	0.03	*0.76*
GG	98	−0.11	*0.44*	−0.16	*0.20*	−0.10	*0.40*	−0.02	*0.88*	−0.005	*0.96*
**SNP28**											
GG	458	-	*-*	-	*-*	-	*-*	-	*-*	-	*-*
AG	76	0.07	*0.61*	0.13	*0.26*	0.16	*0.17*	−0.03	*0.81*	−0.06	*0.62*
AA	4	0.16	*0.75*	0.26	*0.59*	0.34	*0.49*	−0.10	*0.84*	−0.69	*0.12*
**SNP38**											
GG	428	-	*-*	-	*-*	-	*-*	-	*-*	-	*-*
CG	96	0.05	*0.71*	0.11	*0.31*	0.16	*0.67*	0.01	*0.94*	−0.02	*0.86*
CC	4	0.16	*0.75*	0.28	*0.55*	0.17	*0.13*	−0.09	*0.86*	−0.68	*0.13*
**SNP40**											
CC	273	-	*-*	-	*-*	-	*-*	-	*-*	-	*-*
CT	226	−0.09	*0.38*	−0.07	*0.40*	−0.004	*0.97*	−0.04	*0.70*	−0.14	*0.09*
TT	35	−0.03	*0.88*	0.12	*0.48*	−0.08	*0.65*	−0.17	*0.36*	−0.28	*0.09*
**SNP45**											
GG	199	-	*-*	-	*-*	-	*-*	-	*-*	-	*-*
AG	252	−0.16	*0.11*	0.01	*0.91*	0.001	*0.10*	0.000	*0.99*	−0.09	*0.32*
AA	77	−0.27	*0.06*	0.08	*0.53*	0.000	*0.99*	−0.12	*0.39*	−0.08	*0.50*
**SNP49**											
AA	135	-	*-*	-	*-*	-	*-*	-	*-*	-	*-*
AG	284	0.11	*0.31*	0.04	*0.67*	0.04	*0.73*	0.08	*0.46*	0.07	*0.46*
GG	117	0.26	*0.06*	−0.07	*0.57*	−0.10	*0.41*	0.08	*0.58*	0.20	*0.08*

^1^ Data are analyzed using linear regression analysis, adjusted for age, sex, and glucose metabolism status (impaired glucose metabolism and type 2 diabetes as dummy variables with normal glucose metabolism as reference). The genotype with the highest frequency was used as reference, with the other two genotypes added as dummy variables. All outcome variables were standardized and, apart from fasting plasma MGO, all outcome variables were ln-transformed prior to standardization. ^2^
*GLO1* mRNA expression as measured in white blood cells, expressed as ratio of Glo1 versus reference genes. GLO1, glyoxalase-1; iAUC MGO, incremental area under the curve of MGO after an OGTT; MGO, methylglyoxal; OGTT, oral glucose tolerance test; SNP, single nucleotide polymorphism.

**Table 3 antioxidants-10-00219-t003:** Association of GLO1 SNPs with MGO-derived AGEs in plasma and urine^1^.

		Plasma						Urine			
		Free CEL		Free MG-H1		PB CEL		Free CEL		Free MG-H1	
	N	β	*p*	β	*p*	β	*p*	β	*p*	β	*p*
**SNP4**											
AA	423	-	*-*					-	*-*	-	*-*
AT	102	0.16	*0.14*	−0.07	*0.50*	0.17	*0.13*	0.15	*0.19*	−0.06	*0.60*
TT	7	0.16	*0.67*	−0.10	*0.80*	−0.35	*0.37*	0.13	*0.73*	−0.06	*0.87*
**SNP13**											
GG	148	-	*-*	-	*-*	-	*-*	-	*-*	-	*-*
AG	249	0.05	*0.62*	0.003	*0.97*	−0.008	*0.94*	−0.004	*0.97*	−0.006	*0.96*
AA	126	0.10	*0.39*	0.07	*0.53*	0.003	*0.98*	−0.09	*0.49*	−0.10	*0.40*
**SNP16**											
GG	397	-	*-*	-	*-*	-	*-*	-	*-*	-	*-*
AG	127	0.11	*0.27*	0.12	*0.24*	0.01	*0.90*	0.06	*0.57*	0.03	*0.77*
AA	11	0.10	*0.73*	0.20	*0.50*	0.09	*0.78*	−0.47	*0.16*	−0.12	*0.71*
**SNP18**											
TT	161	-	*-*	-	*-*	-	*-*	-	*-*	-	*-*
GT	277	−0.12	*0.22*	−0.02	*0.81*	−0.11	*0.27*	0.02	*0.82*	0.03	*0.74*
GG	98	−0.12	*0.33*	−0.02	*0.87*	0.008	*0.95*	0.08	*0.57*	0.10	*0.44*
**SNP28**											
GG	458	-	*-*	-	*-*	-	*-*	-	*-*	-	*-*
AG	76	0.12	*0.32*	0.14	*0.26*	0.04	*0.76*	−0.01	*0.97*	0.03	*0.81*
AA	4	0.32	*0.51*	0.22	*0.66*	0.11	*0.82*	−0.40	*0.41*	−0.20	*0.68*
**SNP38**											
GG	428	-	*-*	-	*-*	-	*-*	-	*-*	-	*-*
CG	96	0.17	*0.13*	0.12	*0.26*	0.03	*0.81*	0.05	*0.67*	0.05	*0.69*
CC	4	0.32	*0.49*	0.22	*0.65*	0.12	*0.81*	−0.38	*0.43*	−0.20	*0.69*
**SNP40**											
CC	273	-	*-*	-	*-*	-	*-*	-	*-*	-	*-*
CT	226	−0.06	*0.50*	−0.03	*0.77*	−0.06	*0.48*	0.02	*0.83*	−0.03	*0.76*
TT	35	0.05	*0.79*	0.14	*0.43*	−0.03	*0.88*	0.007	*0.97*	0.10	*0.57*
**SNP45**											
GG	199	-	*-*	-	*-*	-	*-*	-	*-*	-	*-*
AG	252	−0.09	*0.36*	−0.10	*0.29*	0.01	*0.88*	−0.01	*0.88*	−0.10	*0.30*
AA	77	0.17	*0.21*	0.003	*0.98*	0.11	*0.42*	0.13	*0.33*	−0.05	*0.71*
**SNP49**											
AA	135	-	*-*	-	*-*	-	*-*	-	*-*	-	*-*
AG	284	−0.24	*0.02*	−0.11	*0.26*	−0.03	*0.78*	−0.12	*0.27*	−0.03	*0.80*
GG	117	−0.15	*0.23*	−0.05	*0.71*	−0.05	*0.69*	−0.08	*0.55*	0.04	*0.75*

^1^ Data are analyzed using linear regression analysis, adjusted for age, sex, and glucose metabolism status (impaired glucose metabolism and type 2 diabetes as dummy variables with normal glucose metabolism as reference). The genotype with the highest frequency was used as reference, with the other two genotypes added as dummy variables. All outcome variables were standardized and, apart from fasting plasma MGO, all outcome variables were ln-transformed prior to standardization. AGEs, advanced glycation endproducts; CEL, Nε-(1-carboxyethyl)lysine; GLO1, glyoxalase-1; MG-H1, Nδ-(5-hydro-5-methyl-4-imidazolon-2-yl)-ornithine; SNP, single nucleotide polymorphism.

## Data Availability

The datasets analyzed during the current study are available from the corresponding author on reasonable request.

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
