# Peer review of "Polymorphisms in Glyoxalase I Gene Are Not Associated with Glyoxalase I Expression in Whole Blood or Markers of Methylglyoxal Stress: The CODAM Study"

_antioxidants, 2021, doi:10.3390/antiox10020219_

Round 1

Reviewer 1 Report

In the explorative study by Maasen et al., the association between nine common single nucleotide polymorphisms (SNPs) identified in the gene GLO1 and the expression of the same gene in white blood cells was examined. In addition, the association between these SNPs and the severity of MGP-stress was also assessed using the levels of multiple markers of MGO-stress in urine. The data were from the Cohort on Diabetes and Atherosclerosis Maastricht (CODAM) study.

Overall, this study was well designed with analyses logically conducted. The methods employed were appropriate and results properly interpreted. Although the negative findings, this manuscript does add new knowledge to diabetic-associated literature and is of interests to the readers of this journal. However, the reviewer does have some concerns as outlined below.

Major concerns:

  1. This study aimed to correlate the SNP polymorphism in the GLO1 gene with its mRNA levels, and the enzymatic products of GLO1. The reviewer consider the protein level of the enzyme (GLO1) and enzyme activity also important parameters, and therefore should be included. It would be helpful if the authors provide some perspective of why these measurements were not included in the study.
  2. The authors should provide the rationale why white blood cells were selected as the source of data collection.
  3. It was an intriguing finding that GLO1 mRNA level was positively correlated with concentrations of its substrate, MGO, in fasting plasma and urine samples (page 5, line 196-198). It is helpful if the authors can speculate the possible causes for this unusual observation.

Minor concerns:

The manuscript was overall well written, clear and easy to follow. Just a few minor issues for consideration:

1.The gene symbol of human glyoxalase 1 should be GLO1 and GLO1 mRNA, according to the Journal of Clinical Investigation guide on gene nomenclature and style.

2. Some abbreviations were provided with full names in the main text, some were provided in both main text and figure legends, some were only provided in the figure legend, but some was not provided at all, such as OGTT. A consistence is recommended.

3. On page 5, lines 199-202, the second sentence was a repeat.

Author Response

Response to Reviewer 1 comments:

Major concerns:

  1. This study aimed to correlate the SNP polymorphism in the GLO1 gene with its mRNA levels, and the enzymatic products of GLO1. The reviewer consider the protein level of the enzyme (GLO1) and enzyme activity also important parameters, and therefore should be included. It would be helpful if the authors provide some perspective of why these measurements were not included in the study.

Response 1: We thank the reviewer for this question. We indeed focused on the associations of SNPs in the GLO1 gene with GLO1 mRNA and the enzymatic substrates and products of glo1. These data were available in our cohort, whereas protein level of glo1 and enzyme activity were not available. We agree with the reviewer that this are also interesting parameters, but measuring them requires much more elaborate techniques. Because glo1 is an intracellular protein, measuring glo1 protein and activity would require isolation of white blood cells or biopsies, and these materials are not available in  our human cohort. Alternatively, glo1 activity can be measured in plasma, but this is most probably mainly a representation of leakage of proteins from cells into the plasma after cell lysis. Therefore, plasma glo1 activity is not an accurate representation of intracellular enzyme activity, and not considered very informative. Therefore, we decided to focus on the expression, and the substrates and products of the glo1 enzyme as outcomes, which can be considered as a reflection of glo1 enzyme functioning.

    2. The authors should provide the rationale why white blood cells were selected as the source of data collection.

Response 2: The main focus of our study was to examine whether SNPs in the GLO1 gene are associated with expression of the GLO1 gene and plasma concentrations of substrates and products of the glo1 enzyme, and thus whether these SNPs are likely functional SNPs. We expect that GLO1 mRNA alters the ability of the glo1 enzyme to detoxify MGO into D-lactate, and therefore that MGO and D-lactate reflect the functionality of the glo1 enzyme. Both the genotyping of the GLO1 gene, as well as the measurement of substrates and products of the glo1 enzyme were performed in peripheral blood samples. In human cohorts it is important to use non-invasive sample collection when possible, such as the collection of peripheral blood samples. Other tissues were not available in this cohort. Therefore, we considered it a valid approach to include GLO1 mRNA expression measured in blood samples.   

  1. It was an intriguing finding that GLO1 mRNA level was positively correlated with concentrations of its substrate, MGO, in fasting plasma and urine samples (page 5, line 196-198). It is helpful if the authors can speculate the possible causes for this unusual observation.

Response 3: We agree with the reviewer that the positive correlation observed between GLO1 mRNA and MGO was an intriguing finding. However, this correlation has to be interpreted with care, because the samples in which GLO1 mRNA was measured were collected seven years after the samples in which the other outcomes, including MGO concentrations in fasting plasma and urine, were measured. This is also described in the methods section (lines 72-73 of the revised manuscript). We have now also included this point of concern in the interpretation of this correlation in the discussion: “Given  the seven year difference between sample collection for measurement of Glo1 expression and measurement of the other outcomes, these correlations have to be interpreted with care, but a possible underlying mechanism is the upregulation of Glo1 as a response to elevated MGO concentrations, via KEAP1 modification and subsequent Nrf1 activation, which regulates Glo1 transcription [40].” (lines 350-354 of revised manuscript). Nevertheless, in this section in the discussion, we have discussed the possible underlying mechanisms how elevated MGO concentrations can lead to elevated GLO1 mRNA; i.e. an MGO modification of  KEAP1, subsequent Nrf1 activation, and  Glo1 transcription. Besides this interpretation already present in the discussion, we have now also emphasized the time difference between the GLO1 mRNA and MGO measurements more in the results section of the revised manuscript: “Glo1 mRNA expression was positively correlated with plasma and urinary concentrations of D-lactate, the product of MGO detoxification (rho=0.2). Glo1 mRNA expression was also positively correlated with concentrations of its substrate, MGO, in fasting plasma (rho=0.2), as well as iAUC MGO and MGO-derived AGEs in urine (rho=0.1). Notably, GLO1 mRNA was measured in samples collected at follow-up, seven years after measurements of MGO and D-lactate.” (lines 196-200).

Minor concerns:

The manuscript was overall well written, clear and easy to follow. Just a few minor issues for consideration:

1.The gene symbol of human glyoxalase 1 should be GLO1 and GLO1 mRNA, according to the Journal of Clinical Investigation guide on gene nomenclature and style.

Response 1: We have adapted this throughout the manuscript.

  1. Some abbreviations were provided with full names in the main text, some were provided in both main text and figure legends, some were only provided in the figure legend, but some was not provided at all, such as OGTT. A consistence is recommended.

Response 2: Thank you for noticing this. We have adapted this throughout the manuscript.

  1. On page 5, lines 199-202, the second sentence was a repeat.

Response 3: Thank you for your comment. The first sentence described the reciprocal correlation between the AGEs CEL and MG-H1 in plasma. The second sentence described the correlation between CEL measured in plasma and urine, and the correlation between MG-H1 measured in plasma and in urine. We agree with the reviewer that this was not clear, and have therefore adapted this section to the following: “We observed a strong reciprocal correlation between the free forms of the MGO-derived AGEs, CEL and MG-H1, measured in plasma (rho=0.7), as well as between CEL and MG-H1 measured in urine (r=0.6). Also, we observed a strong correlation between free CEL measured in plasma and in urine, and the same holds true for free MG-H1 in plasma and urine (both rho=0.7)(lines 201-205 revised manuscript).

Reviewer 2 Report

The manuscript by Maaseen et al. evaluates the potential association of SNPS in GLO1 and detoxifcation markers of methylglyoxal using data from the the Cohort on Diabetes and Atherosclerosis Maas-17 tricht (CODAM). The authors used markers of methylglyoxal in plasma and urine along with the expression of GLO1. The study presents different limitations, including that it was performed in a unique ethnical group (Caucasian) with a reduced small size and only used white blood cells for the analysis. It would desirable a similar analysis in a different cohort to validate the findings. In spite of that, the study is well-designed, the manuscript is well-written and the conclusion drawn are adequate.

The authors found null findings: no association between any polymorphism in GLO1 and any of the parameters analyzed in the study (Glo1 mRNA expression and markers of MGO stress). It suggests that the SNPs probably do not play a major a major impact in the detoxification of methylglyoxal. This study shed light in the field where there is not a consensus given that previous reports are inconsistent about the potential impact of these SNPs. For all these reasons, I would recommend the publication of this research.

Author Response

We thank the reviewer for the efforts put into reviewing our manuscript, the kind words, and for recommending our manuscript for publication.